

# Tracking the virus-like particles of *Macrobrachium rosenbergii* nodavirus in insect cells

Ummi Fairuz Hanapi[1], Chean Yeah Yong[1], Zee Hong Goh[1], Noorjahan Banu Alitheen[2,3], Swee Keong Yeap[4] and Wen Siang Tan[1,3]

[1] Department of Microbiology, Faculty of Biotechnology and Biomolecular Sciences, Universiti Putra Malaysia, Serdang, Selangor, Malaysia

[2] Department of Cell and Molecular Biology, Faculty of Biotechnology and Biomolecular Sciences, Universiti Putra Malaysia, Serdang, Selangor, Malaysia

[3] Institute of Bioscience, Universiti Putra Malaysia, Serdang, Selangor, Malaysia

[4] Xiamen University Malaysia, Sepang, Selangor, Malaysia

Corresponding author
Wen Siang Tan, wstan@upm.edu.my, wensiangtan@yahoo.com

## ABSTRACT

*Macrobrachium rosenbergii* nodavirus (*Mr*Nv) poses a major threat to the prawn industry. Currently, no effective vaccine and treatment are available to prevent the spread of *Mr*Nv. Its infection mechanism and localisation in a host cell are also not well characterised. The *Mr*Nv capsid protein (*Mr*Nvc) produced in *Escherichia coli* self-assembled into virus-like particles (VLPs) resembling the native virus. Thus, fluorescein labelled *Mr*Nvc VLPs were employed as a model to study the virus entry and localisation in *Spodoptera frugiperda*, Sf9 cells. Through fluorescence microscopy and sub-cellular fractionation, the *Mr*Nvc was shown to enter Sf9 cells, and eventually arrived at the nucleus. The presence of *Mr*Nvc within the cytoplasm and nucleus of Sf9 cells was further confirmed by the Z-stack imaging. The presence of ammonium chloride ($NH_4Cl$), genistein, methyl-$\beta$-cyclodextrin or chlorpromazine (CPZ) inhibited the entry of *Mr*Nvc into Sf9 cells, but cytochalasin D did not inhibit this process. This suggests that the internalisation of *Mr*Nvc VLPs is facilitated by caveolae- and clathrin-mediated endocytosis. The whole internalisation process of *Mr*Nvc VLPs into a Sf9 cell was recorded with live cell imaging. We have also identified a potential nuclear localisation signal (NLS) of *Mr*Nvc through deletion mutagenesis and verified by classical-NLS mapping. Overall, this study provides an insight into the journey of *Mr*Nvc VLPs in insect cells.

# INTRODUCTION

*Macrobrachium rosenbergii* or commonly known as the giant freshwater prawn is commercially grown in Asia, the Western Pacific Islands and South America (*Pillai & Bonami, 2012*). However, the white tail disease (WTD) caused by *Macrobrachium rosenbergii* nodavirus (*Mr*Nv) is currently the biggest problem that devastates giant freshwater prawn farming, which always results in 100% mortality (*Qian et al., 2003*; *Nair & Salin, 2012*). Post-larvae
diagnosed with WTD have a whitish colouration on the abdominal and tail segments as well as muscle (*Bonami & Widada, 2011*). Post-larvae of *M. rosenbergii* are more susceptible to *Mr*Nv infection and adult prawns are believed to be the virus carriers (*Hayakijkosol & Owens, 2013*).

*Mr*Nv is classified in the family *Nodaviridae*. Being rather distinctive to other nodaviruses, *Mr*Nv has been proposed to be placed into a new genus, *Gammanodavirus* (*Naveenkumar et al., 2013*). The virus particle has an icosahedral and non-enveloped capsid of about 27 nm in diameter (*Qian et al., 2003*), encapsidating bipartite single stranded RNAs: RNA1 of about 2.9 kb encoding the RNA-dependent RNA polymerase; RNA2 of approximately 1.26 kb encodes the capsid protein. The *capsid* gene has been cloned into a bacteria plasmid and the recombinant *Mr*Nv capsid protein (*Mr*Nvc) self-assembled into virus-like particles (VLPs), resembling the native *Mr*Nv isolated from infected prawns (*Goh et al., 2011*). By studying these VLPs, the RNA binding domain of *Mr*Nv was located at amino acids 20–29 of the capsid protein (*Goh et al., 2014*). Therefore, *Mr*Nvc VLPs provide an alternative method to study the virus structure and its functions in host cells.

The entry, trafficking and localisation of nodaviruses in their host cells were obtained mainly through the study of Flock House virus (FHV), an *Alphanodavirus*. FHV enters *Drosophila melanogaster* cells through receptor-mediated endocytosis which requires an acidic condition. Pre-treatment of cells with $NH_4Cl$ and bafilomycin A1 prevented acidification of endosomes and inhibited FHV's infection (*Odegard, Banerjee & Johnson, 2010*). Under normal condition, internalised FHV is enclosed in an acidic endosome. The acidic pH in the endosomal compartment triggers conformational changes of the viral capsid proteins which expose and release the proteolytically cleaved 4.4 kDa gamma ($\gamma$) peptides. The particle associated $\gamma$ peptides then disrupt the endosomal membrane to facilitate the release of viral RNAs and nucleocapsid into the cytoplasm (*Odegard, Banerjee & Johnson, 2010*).

FHV does not translocate into nucleus. On the other hand, greasy grouper nervous necrosis virus (GGNNV), a *Betanodavirus*, was found not only in the cell cytoplasm, but also in the nucleolus (*Guo, Dallmann & Kwang, 2003*). A 9-residue peptide ([23]RRRANNRRR[31]) located at the N-terminal region of the viral capsid protein and highly rich in positively charged amino acids, was identified as a nucleolus targeting sequence of GGNNV. This 9-residue peptide shares some similarities with the RNA-binding region of *Mr*Nvc ([20]KRRKRSRRNR[29]) located at the N-terminal end of the viral capsid protein (*Goh et al., 2014*). This indicates that the RNA-binding region of *Mr*Nvc could be the nucleolus targeting sequence.

Recently, *Somrit et al. (2016)* successfully infected *Spodoptera frugiperda*, Sf9 cells with *Mr*Nv. They revealed that the *Mr*Nv infected Sf9 cells through a caveolae-mediated endocytosis pathway. The main aim of the current study was to trace the trafficking of *Mr*Nvc VLPs in Sf9 cells and compare with that of the live virus. Thus, the *Mr*Nvc VLPs were labelled with NHS-fluorescein and their movement from the cell surface to the nucleus was studied with fluorescence microscopy, live cell imaging system and sub-cellular fractionation. This is the first report on the trafficking of *Mr*Nvc VLPs in Sf9 cells.

## MATERIALS AND METHODS

### Purification of *Mr*Nvc VLPs

Expression and purification of the *Mr*Nvc and its N-terminal deletion mutants (9Δ*Mr*Nvc, 19Δ*Mr*Nvc, 29Δ*Mr*Nvc and 20–29Δ*Mr*Nvc) were as described previously by *Goh et al. (2011)* and *Goh et al. (2014)*. Briefly, *E. coli* cells harbouring the recombinant plasmids were grown in Luria-Bertani broth (500 ml) containing ampicillin (50 mg/ml) at 220 rpm for overnight. *E. coli* cultures were induced for recombinant protein expression with IPTG (1 mM) at 37 °C for 5 h. Cells were then pelleted and lysed in lysis buffer (25 mM HEPES, 500 mM NaCl; pH 7.4) by adding phenylmethylsulfonyl fluoride (PMSF, 2 mM), $MgCl_2$ (4 mM), freshly prepared lysozyme (0.2 mg/ml) and DNase 1 (0.02 mg/ml). After 2 h of incubation at room temperature (RT), the cells were sonicated at 200 Hz, 15 times with 15 s interval. The mixture was centrifuged at $10,000 \times$ g and supernatant was loaded into HisTrap HP columns (1 ml; GE Healthcare, Buckinghamshire, United Kingdom). Washing buffer A (25 mM HEPES, 500 mM NaCl, 50 mM imidazole; pH 7.4) and B (25 mM HEPES, 500 mM NaCl, 200 mM imidazole; pH 7.4) were used to wash the unbound proteins. Elution buffer (25 mM HEPES, 500 mM NaCl, 500 mM imidazole; pH 7.4) was used to elute *Mr*Nv recombinant proteins. Eluted capsid proteins were dialysed in HEPES buffer (25 mM HEPES, 150 mM NaCl; pH 7.4) at 4 °C, overnight.

### Fluorescence microscopy

NHS-fluorescein (1 mg; Thermo Scientific, Rockford, USA) was dissolved in DMSO (100 μl). Dialysed *Mr*Nvc VLPs (100 μg) were incubated with NHS-fluorescein solution (0.01 mg/ml) at 4 °C for 24 h and then dialysed in HEPES buffer for 24 h. Sf9 cells (ATCC® CRL-1711™; $8 \times 10^5$ cells) were seeded on glass coverslips in a 6-well culture plate with Sf900 II SFM medium (2 ml; GIBCO, Grand Island, NY, USA) for 24 h and the medium in each well was changed with a fresh medium. The fluorescein labelled VLPs (F-*Mr*Nvc VLPs) were filtered with a membrane filter (0.4 μm) and the filtered VLPs (25 μg) were added into the culture plates in triplicates. Non-labelled *Mr*Nvc VLPs were used as negative controls. The cells were incubated for 16 h at RT to allow maximum nuclear translocation of *Mr*Nvc. The cells in each well were then incubated with the Cell Tracker™ Orange CMTMR Dye (0.5 μM; Thermo Fisher Scientific, Massachusetts, USA) in medium (2 ml) for 30 min to stain the cell cytoplasm. The coverslips were then washed with PBS (137 mM NaCl, 2.7 mM KCl, 4.3 mM $Na_2HPO_4$, 1.47 mM $KH_2PO_4$; pH 7.4) for six times and the cells were fixed with paraformaldehyde (3.7% in PBS; pH 7.4) for 10 min at RT. One drop of NucBlue Live Ready Probe Reagent (Life Technologies, Carlsbad, CA) was added to the mounting medium (1 ml; 90% glycerol, 20 mM Tris–HCl (pH 8.5) and 100 mM propyl gallate) to stain the cell nucleus. A drop of mounting medium was placed on a glass slide and the coverslip was carefully put on top of it. The edge of the coverslip was sealed with a nail polish and the slides were observed under a fluorescence microscope, LEICA DM2500 (Leica Camera, Solms, Germany) with green (excitation filter BP 515–560; emission filter LP 590), blue (excitation filter BP 450–490; emission filter LP 515) and red (excitation filter BP 590–650; emission filter BP 700/75) filters under 40× objective.

Z-stack images were captured through the use of a confocal fluorescence microscope, Olympus IX81 (Olympus, Tokyo, Japan) equipped with Disk Scanning Unit (DSU) for spinning disk confocal (pinhole diameter 50–300 μm (1 μm step)) under 60× objective.

## Sub-cellular fractionation

Fractionation of Sf9 cells was done according to the method described by *Guo, Dallmann & Kwang (2003)* with slight modifications on the buffers used. Confluent Sf9 cells in flasks (75 cm$^2$) were added with medium (10 ml) containing *Mr*Nvc VLPs (1 mg) and incubated for 16 h. The subsequent steps were done at 4 °C. Cells were washed with ice-cold PBS (pH 7.4) for three times. PBS (3 ml; pH 7.4) containing the Protease Inhibitor Cocktail (10 μl; SIGMA Chemical Co., St. Louis, MO, USA) was added before dislodging the cells from the culture flasks with a cell scraper. The cells were harvested by centrifugation at 260× g for 5 min. Hypotonic buffer (20 mM HEPES (pH 7.9), 10 mM KCl, 1 mM EDTA, 0.1 mM Na$_2$HPO$_4$, 20 mM NaF, 10% glycerol, 0.1% tween-20, 1 mM DTT, 1 mM PMSF and 1% Protease Inhibitor Cocktail) was used to resuspend the cell pellets. The suspension was incubated for 30 min on ice. The suspension was vortexed for 1 min and incubated on ice for 1 min, and this process was repeated for five times. The cytoplasmic extract of Sf9 cells was retrieved by centrifugation at 11,330× g at 4 °C for 10 min. The remaining pellet was washed with hypotonic buffer and centrifuged with the same condition. The nuclear pellet was resuspended in hypotonic buffer (without tween-20) with the addition of NaCl (420 mM) and glycerol (20%). The suspension was vortexed as above for 5 times. Then, the nuclear lysate was centrifuged at 11,330× g at 4 °C for 10 min. The supernatant containing nuclear extract was collected for further analysis with SDS-PAGE and Western blotting.

## SDS-PAGE and Western blot analysis

Cytoplasmic and nuclear extracts of Sf9 cells incubated with the VLPs were electrophoresed in SDS-polyacrylamide (12%) gels at 16 mA. The gels were stained with the Coomassie Brilliant Blue (CBB) dye. For Western blotting, the proteins on the gels were transferred onto nitrocellulose membranes and blocked with skimmed milk (10%; Anlene, Auckland, New Zealand) in TBS buffer (50 mM Tris–HCl (pH 7.4) and 150 mM NaCl) for 1 h. The membranes were washed three times with TBST (50 mM Tris–HCl (pH 7.4), 150 mM NaCl and 0.1% (v/v) Tween-20) for 5 min each, and then incubated with anti-His antibody (1:5000 dilution with TBS; GE Healthcare, Buckinghamshire, United Kingdom) or rabbit anti-*Mr*Nv serum (1:1000 dilution) at 4 °C. After an overnight incubation, the membrane was washed again with TBST for five times, 5 min each and incubated with anti-mouse IgG antibody conjugated to alkaline phosphatase (1:5000 dilutions in TBS; KPL Inc., Maryland, USA) or anti-rabbit IgG (1:5000 dilution with TBS; SIGMA Chemical Co., St. Louis, MO, USA) overnight at 4 °C. After washing five times with TBST, the protein bands were developed by adding 5-bromochloroindolyl-phosphate (3.3 μl/ml) and nitrobluetetrazolium (6.6 μl/ml) in alkaline phosphatase buffer (100 mM Tris–HCl (pH 9.5), 100 mM NaCl and 5 mM MgCl$_2$).

## Live cell imaging

Sf9 cells ($8 \times 10^5$ cells) were seeded in 6-well culture plates at RT for overnight. Fresh Sf900 II SFM medium (2 ml) containing F-$Mr$Nvc VLPs (25 µg/ml) was added into the cells and incubated for 1 h at 4 °C, followed by 30 min incubation at RT to initiate internalisation of the $Mr$Nvc VLPs. NucBlue Live Ready Probe Reagent (1 drop) and 3 mM propyl gallate were added to stain the nucleus prior to viewing the uptake of $Mr$Nvc VLPs by Sf9 cells across the plasma membrane in real-time with Olympus IX81-DSU equipped with a high-sensitivity cooled CCD (charge-coupled device) camera with a built-in shutter, which allows single DSU image to be obtained in 0.1–0.4 s and avoid photobleaching when no image is acquired. Fluorescence images were visualised by exciting the fluorescence at 493 nm (green for VLPs) and 360 nm (blue for cell nucleus). Static superimposed images were captured every 30 s for 1 h duration and compiled into a video to observe the uptake of VLPs by the cells.

## Endocytosis inhibition assay

Five endocytosis inhibitors were used to study the uptake mechanism of $Mr$Nvc VLPs into Sf9 cells. The Sf9 cells ($8 \times 10^5$ cells) were seeded on a glass slide in each well of 6-well culture plates for overnight. The Sf900II SFM medium was removed and the cells were pre-incubated in 2 ml medium for 1 h at RT with: cytochalasin D (2 µM; Calbiochem, CA, USA), $NH_4Cl$ (10 mM; Bio Basic Inc., NY, USA), chlorpromazine (CPZ: 50 µM; Santa Cruz Biotechnology, Texas, USA), methyl-$\beta$-cyclodextrin (2 mM; Sigma-Aldrich, MO, USA) and genistein (100 µM; Calbiochem, CA, USA). The cells were then incubated for another 16 h at RT following the addition of F-$Mr$Nvc VLPs (25 µg/ml). The coverslips were washed with PBS (pH 7.4) for six times and the cells were fixed with paraformaldehyde (3.7% in PBS; pH 7.4) for 10 min at RT. A drop of mounting medium was placed on a glass slide and the coverslip was gently placed on top of it. The edge of the coverslip was sealed with a nail polish and the slides were observed under the Zeiss LSM5 PASCAL laser scanning microscope (Carl Zeiss, Oberkochen, Germany) excited with 488 nm laser (pinhole diameter 160 µm) under 63× objective.

## Cell viability assay

The viability of Sf9 cells incubated in different endosomal inhibitors were analysed with 3-(4, 5- dimethylthiazol-2-yl)-2, 5-diphenyltetrazolium bromide (MTT) assay. Sf9 cells (100 µl; $2 \times 10^5$ cells/ml) were seeded on 96-well culture plates and incubated for 16 h at RT in a medium (100 µl) containing cytochalasin D (2 µM), $NH_4Cl$ (10 mM), chlorpromazine (50 µM), methyl-$\beta$-cyclodextrin (2 mM) and genistein (100 µM). Sf9 cells without the inhibitors served as controls. The MTT reagent (10 µl; 5 mg/ml in PBS (pH 7.4)) was added in each well and incubated at RT. After 2 h, the medium was removed and MTT solubilisation solution (100 µl; 1:9 part of 10% SDS:DMSO) was added. The plate was incubated for 15 min in dark and the absorbance at 570 nm was measured with the µQuant™ ELISA plate reader (Bio Tek Instruments, Winooski, USA).

The percentage of viable Sf9 cells was measured as $(100 - ((A - B)/A \times 100))$, where A is the absorbance of the control cells and B is the absorbance of treated cells.

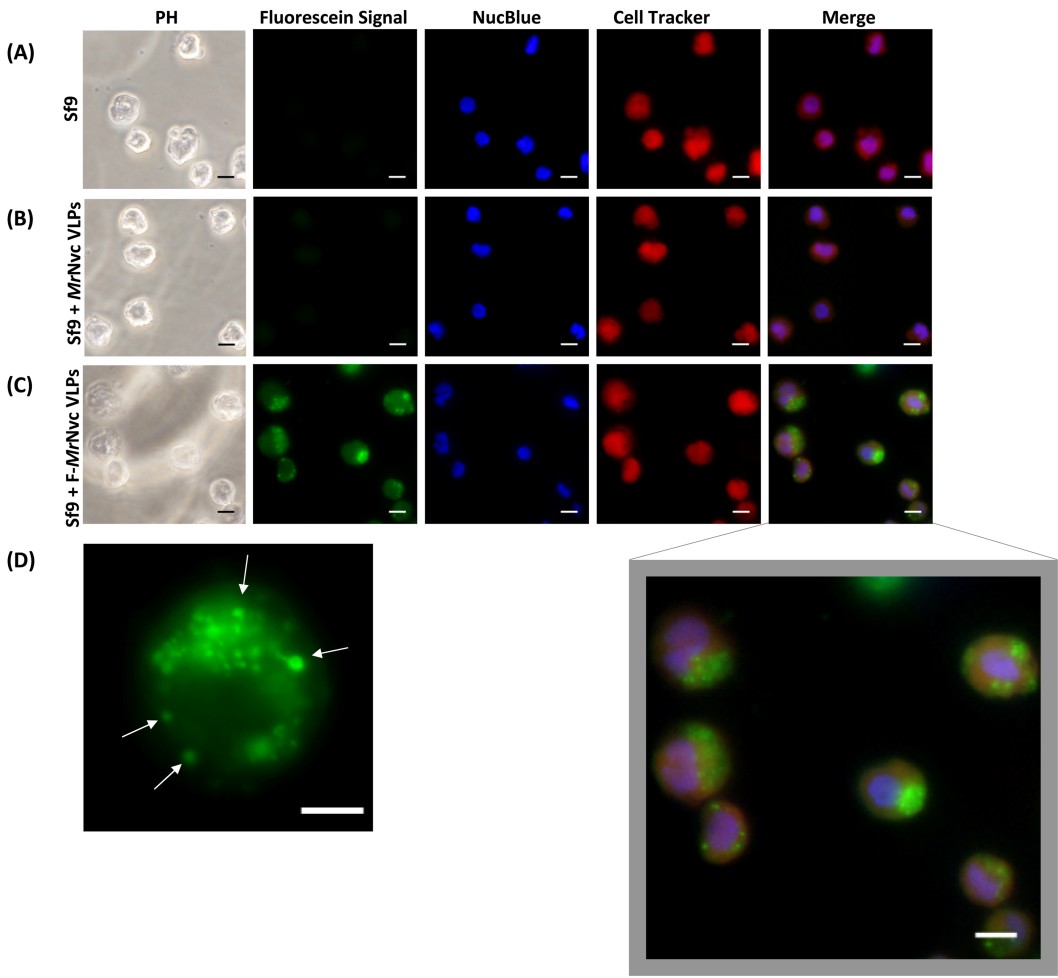

**Figure 1** **Triple fluorescence labelling and detection of *Mr*Nvc VLPs in Sf9 cells.** (A) Sf9 cells in the absence of *Mr*Nvc VLPs, (B) Sf9 cells incubated with non-labelled *Mr*Nvc VLPs and (C) Sf9 cells incubated with F-*Mr*Nvc VLPs. The cell nucleus and cytoplasm were labelled with NucBlue Live Ready Probe Reagent (blue) and Cell Tracker Orange (red), respectively. PH indicates images captured under white light. Merge images represent the superimposed green, blue and red signals. (D) Small granular appearance in a Sf9 cell incubated with F-*Mr*Nvc VLPs is indicated by the arrows. Bars, 10 μm.

## RESULTS

### *Mr*Nvc VLPs internalise Sf9 cells

Fluorescence microscopy was used to examine the delivery and localisation of fluorescein-labelled *Mr*Nvc VLPs (F-*Mr*Nvc VLPs) in Sf9 cells. The VLPs labelled with NHS-fluorescein at the lysine residues of *Mr*Nvc were added to Sf9 cells and incubated at room temperature (Fig. 1). To observe the dispersion of F-*Mr*Nvc VLPs in the Sf9 cells, the NucBlue Live Ready Probe Reagent and Cell Tracker Orange were used to stain the nuclear and cytosolic boundaries, respectively. No intense green fluorescence was detected from the normal Sf9 cells (Fig. 1A) and in the Sf9 cells incubated with non-labelled *Mr*Nvc VLPs (Fig. 1B). After 16 h of incubation with the F-*Mr*Nvc VLPs, green fluorescence accumulated within the Sf9 cells (Fig. 1C), indicating the internalisation of VLPs into the cells. From the merged images,

*Mr*Nvc VLPs were clearly located at the cytoplasm of Sf9 cells. A close-up examination on the cells showed intense fluorescence spots of small granular appearance localised throughout the cell cytoplasm (Fig. 1D). Interestingly, the fluorescence spots tended to cluster at one side of the cells resembling cap-like structures, which have also been reported by *Liu et al. (2005)* when SSN-1 cells were infected with dragon grouper nervous necrosis virus (DGNNV), but not the VLPs.

### *Mr*Nvc distributes in Sf9 cell cytoplasm and nucleus

Sub-cellular fractionation was performed to verify the presence of *Mr*Nvc in Sf9 cells. The cells were incubated with *Mr*Nvc VLPs at 0, 4, 8, 12 and 16 h. At each time point, the cells were disrupted by hypotonic buffer (HB) to fractionate the nuclear and cytoplasmic components. *Mr*Nvc was detected by the rabbit anti-*Mr*Nv serum (Fig. 2A(i)) and anti-His antibody (Fig. 2A(ii)). In the cytoplasmic and nuclear fractions of Sf9 cells incubated with *Mr*Nvc VLPs, a distinct band corresponding to the *Mr*Nvc with a molecular mass ∼46 kDa was detected after 4 h post-incubation. After 8 h of incubation, the *Mr*Nvc was detected in the cell nucleus. In addition, a smaller protein band of ∼44 kDa was detected by the serum and anti-His antibody indicating it is an N-terminal degraded product of *Mr*Nvc. *Goh et al. (2014)* and *Yong et al. (2015a)* also reported that the *Mr*Nvc VLPs produced in *E. coli* contained the N-terminal degraded product. The Sf9 cells incubated with *Mr*Nvc VLPs for 16 h were viewed under a fluorescence microscope and 20 Z-stack images were captured. The orthogonal view of the cells was obtained by using the ImageJ software. From the view of XY, YZ and XZ axes, it was confirmed that the *Mr*Nvc VLPs, as indicated by the yellow line interceptions, were localised in the cell nucleus (Fig. 2B) and cytoplasm (Fig. 2C).

### Effect of endosomal inhibitors on the entry of *Mr*Nvc VLPs into Sf9 cells

$NH_4Cl$ (10 mM), cytochalasin D (2 μM), methyl-$\beta$-cyclodextrin (2 mM), CPZ (50 μM) and genistein (100 μM) were used to study the entry mechanism of *Mr*Nvc VLPs in Sf9 cells. Sf9 cells were pre-incubated with the inhibitors, followed by the addition of F-*Mr*Nvc VLPs. After 16 h, the cells were observed under a fluorescence microscope. The amount of green granular appearance in Sf9 cells decreased drastically in the presence of $NH_4Cl$, CPZ, methyl-$\beta$-cyclodextrin and genistein (Fig. 3A(iv–vii)). These results suggest that $NH_4Cl$, methyl-$\beta$-cyclodextrin, CPZ and genistein inhibited *Mr*Nvc VLPs entry at the early event of endosomal pathway in Sf9 cells. On the other hand, cytochalasin D did not inhibit the VLPs entry (Fig. 3A(iii)).

MTT assay was performed to assess Sf9 cell viability in the presence of $NH_4Cl$, cytochalasin D, CPZ, methyl-$\beta$-cyclodextrin and genistein. All treated cells have viability percentage over 80% (Fig. 3B), indicating the treatments were not toxic to the cells.

### The trafficking mechanism of *Mr*Nvc VLPs into Sf9 cells examined with live cell imaging system

The internalisation process of *Mr*Nvc VLPs into Sf9 cells was examined using a live cell imaging system. After 1 h pre-incubation at 4 °C, the Sf9 cells were incubated at RT for 30 min. Time lapse video was captured for 1 h in 30 s interval (Video S1). As demonstrated in

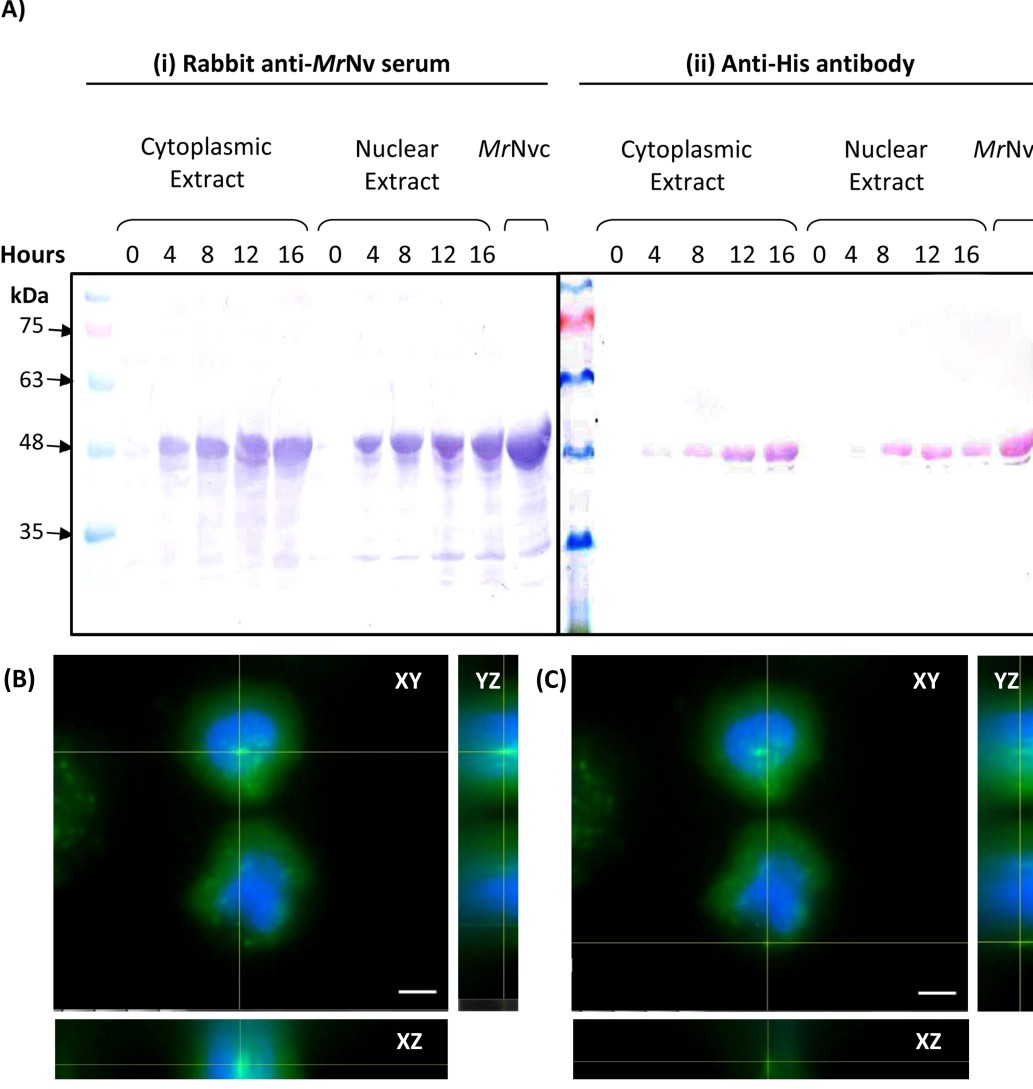

**Figure 2** *Mr*Nvc distribution in Sf9 cytoplasmic and nuclear components. (A) Western blot analysis of the cytoplasmic and nuclear components of Sf9 cells incubated with *Mr*Nvc VLPs (25 μg/ml) at different time (0, 4, 8, 12 and 16 h). Both cytoplasmic and nuclear extracts were probed with rabbit anti-*Mr*Nv serum (A(i)) and anti-His antibody (A(ii)). (B–C) The orthogonal view from the Z-stack images of the green fluorescence in the nucleus (B) and cytoplasm (C) of Sf9 cells incubated with *Mr*Nvc VLPs. Blue fluorescence indicates the cell nucleus. Yellow lines interception shows the location of the green fluorescent spots (F-*Mr*Nvc VLPs). Bars, 5 μm.

Fig. 4A, VLPs attached on the cell surface were brought near to a membrane pit, which then accumulated into the middle of the pit (Fig. 4B). The VLPs were enclosed in the endosomes which move freely in the cytosol (Fig. 4C), until the shape became disproportionate (Fig. 4D). The endosome disappeared completely after a few minutes (Fig. 4E as annotated by the red arrow), in which the *Mr*Nvc is believed to be released into the cytosol. Endosome formation and endosomal escape are summarised in a schematic diagram (Fig. 4F).

Peer J

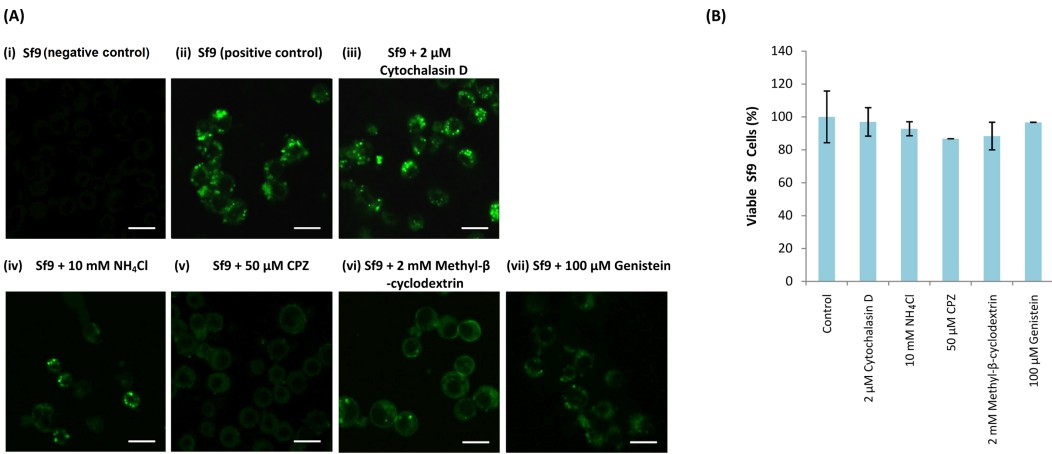

**Figure 3** **Effect of endosomal inhibitors on the entry of _Mr_Nvc VLPs into Sf9 cells.** Sf9 cells were pre-incubated with different endosomal inhibitors: (A(iii)) cytochalasin D (2 μM), (A(iv)) NH$_4$Cl (10 mM), (A(v)) CPZ (50 μM), (A(vi)) methyl-β-cyclodextrin (2 mM) and (A(vii)) genistein (100 μM). _Mr_Nvc VLPs labelled with NHS-fluorescein (F-_Mr_Nvc VLPs; 25 mg/ml) were added to each pre-treated sample and incubated for 16 h in the presence of endosomal inhibitors. Bars, 20 μm. (A(ii)) Sf9 cells added with F-_Mr_Nvc VLPs but without any inhibitor serve as positive control, whereas (A(i)) Sf9 cells without any inhibitor nor F-_Mr_Nvc VLPs serve as negative control. (B) MTT assay showing the viability of Sf9 cells in the presence of inhibitors.

## Incubation of Sf9 cells with the N-terminal deletion mutants of _Mr_Nvc

The N-terminal region of _Mr_Nvc is highly rich in positively charged residues (_Goh et al., 2014_). The roles of this region in cell entry and internalisation into nucleus were studied by four N-terminally deleted mutants (Fig. 5A); namely 9Δ_Mr_Nvc, 19Δ_Mr_Nvc, 29Δ_Mr_Nvc and 20–29Δ_Mr_Nvc. The Sf9 cells incubated with the four deletion mutants and the full length _Mr_Nvc were subjected to sub-cellular fractionation (Fig. 5B) and viewed under a fluorescence microscope with two labelling signals (Fig. 5C). Figure 5B shows that the full length _Mr_Nvc and the four mutants were detected in the cell cytoplasm by Western blotting. The green signal observed under fluorescence microscopy revealed that the fluorescein-labelled VLPs of all mutants and the full length capsid protein appear as green granules in the cell cytoplasm. This study confirmed the Western blot results demonstrating that the N-terminal region of the _Mr_Nvc does not play a role in the entry of VLPs into Sf9 cells, and that the receptor binding site is located somewhere after amino acid 29 of _Mr_Nvc. Deletion of amino acids 20–29 (20–29Δ_Mr_Nvc) significantly decreased the internalisation of _Mr_Nvc into nucleus (Fig. 5B), as no trace of 20–29Δ_Mr_Nvc can be detected in the nuclear extract (analysed with the Quantity One Software), demonstrating the importance of these positively charged residues in nuclear localisation.

## DISCUSSION

_Mr_Nvc produced in _E. coli_ self-assembles into VLPs resembling the native virus isolated from infected prawns (_Goh et al., 2011_). These VLPs have been used in a wide variety of studies, including a fundamental study which has led to the discovery of the RNA-binding region in _Mr_Nvc (_Goh et al., 2014_), and their applications as nano-particles for the delivery

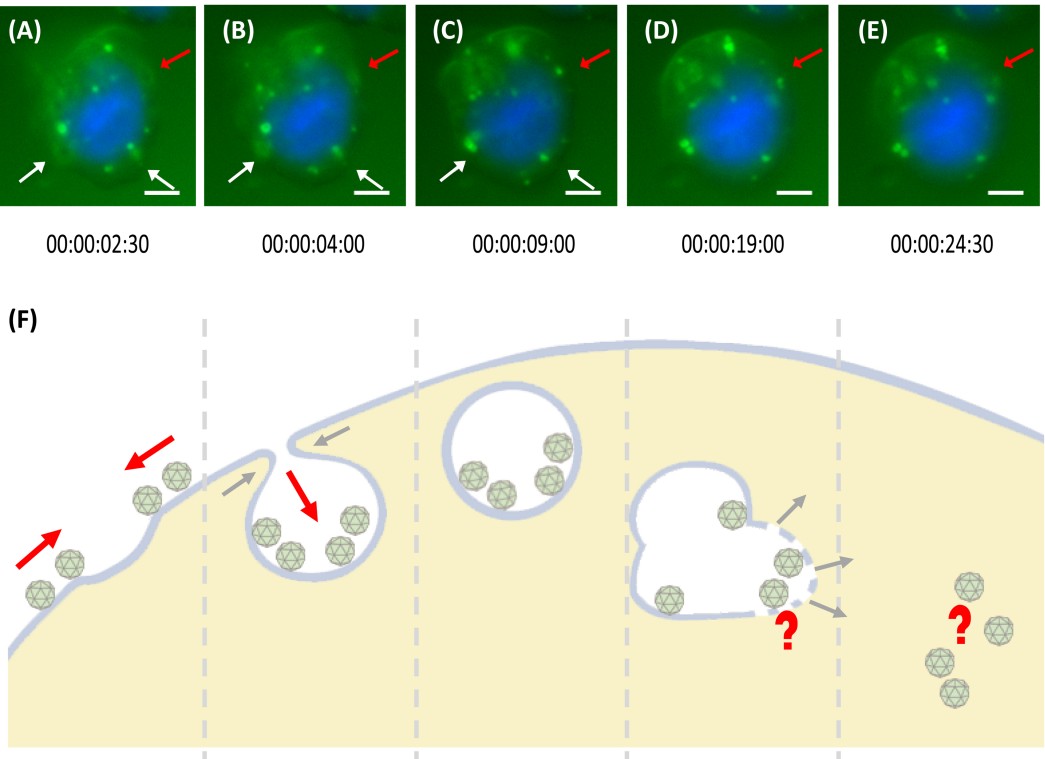

**Figure 4** **Trafficking mechanism of *Mr*Nvc VLPs into Sf9 cells.** (A–E) Live cell imaging of the formation of endosomes and endosomal escape of F-*Mr*Nvc VLPs. Sf9 cells were incubated with F-*Mr*Nvc VLPs at 4 °C for 1 h and shifted to RT for 30 min before they were viewed under a live cell imaging system. Each image was captured in 30 s time lapse for 1 h (Video S1). White arrows indicate endosome formation and red arrows show endosome formation to endosomal escape of VLPs. The live cell imaging image shows (A) the attached F-*Mr*Nvc VLPs were gathered around a hollow membrane pit, and (B) accumulated inside the pit. (C) Endosome enclosing the VLPs was formed. (D) The size and shape of the endosome become disproportionate and (E) F-*Mr*Nvc was released into the cytosol. These images were captured at the specified duration of time lapse (Video S1). (F) A schematic diagram summarising the whole process. Bars, (A–E) 5 μm.

of DNA into insect cells (*Jariyapong et al., 2014*), as well as the display of foreign epitopes such as those of hepatitis B (*Yong et al., 2015a*) and influenza A (*Yong et al., 2015b*) viruses. As such, learning the *Mr*Nvc VLPs' mode of entry into cells will surely benefit its potential applications, escpecially as a gene or drug delivery vehicle. Due to the ease of production and manipulation, the *Mr*Nvc VLPs serve as an excellent model to study the *Mr*Nv trafficking within host cells. Most recently, *Mr*Nv has been shown to infect Sf9 cells (*Somrit et al., 2016*). Therefore, in this study, the *Mr*Nvc VLPs produced in *E. coli* were labelled with fluorescein and their localisation in Sf9 cells was studied with fluorescence microscopy, sub-cellular fractionation and live cell imaging system.

    *Mr*Nv was reported to infect *Macrobrachium rosenbergii* (*Hameed & Yoganandhan, 2004*) and *Penaeus vannamei* (*Tang et al., 2007*). In this study, we have demonstrated the ability of *Mr*Nvc VLPs to internalise Sf9 cells. This suggests that the Sf9 cells and prawn cells share similar receptor for the binding of *Mr*Nv. Upon entry of *Mr*Nvc VLPs, granules

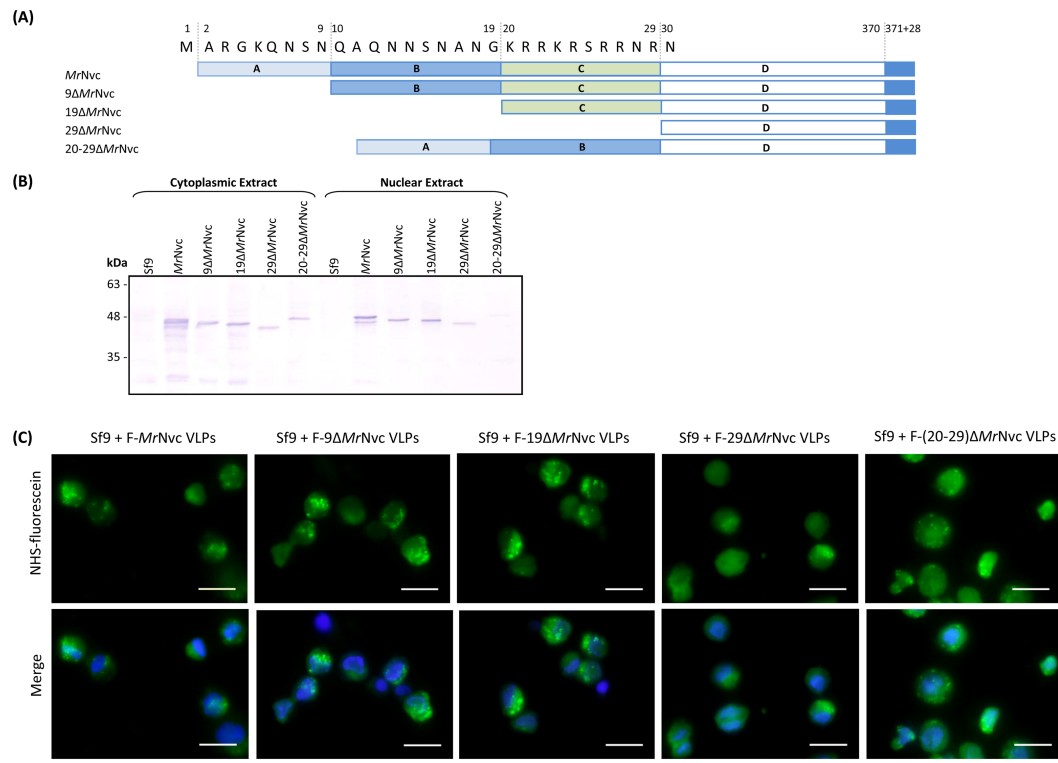

**Figure 5** **Sub-cellular localisation of the N-terminal deletion mutants of *Mr*Nvc in Sf9 cells** (A) Four mutants and the full length *Mr*Nvc were used to infect Sf9 cells for 16 h. (B) The cytoplasm and nuclear components were extracted and analysed with Western blotting by using rabbit anti-*Mr*Nv serum to examine the localisation sites. Sf9 cells served as negative controls. (C) The N-terminal mutants and the full length capsid proteins were labelled with NHS-fluorescein and viewed under a fluorescence microscope. Blue colour indicates the cell nucleus. The merged images are the superimposed images of blue and green signals. Bars, 20 μm.

emitting green fluorescence were observed in the cytoplasm of Sf9 cells under a fluorescence microscope. These green granules are believed to be clusters of VLPs entrapped in endosomes.

In the effort to understand better the mechanism of virus uptake, $NH_4Cl$ was used to disrupt the function of endosome and its role in *Mr*Nvc VLPs uptake. $NH_4Cl$ raises the endocytic pH and inhibits endosomal acidification. The appearance of green fluorescence spots in Sf9 cells incubated with *Mr*Nvc VLPs reduced significantly in the presence of $NH_4Cl$ (10 mM). This suggests that the *Mr*Nvc VLPs entered host cell via acidic endocytic pathway involving the formation of endosomes. Treatment of Sf9 cells with genistein, which inhibits caveolae-mediated endocytosis by inhibiting several tyrosine kinase functions in cells (*Iversen, Skotland & Sandvig, 2011*), has reduced the uptake of *Mr*Nvc VLPs by Sf9 cells dramatically. This corresponded well with the results reported by *Somrit et al. (2016)*, who demonstrated that in the presence of genistein, *Mr*NV infection in Sf9 cells was reduced significantly. In addition, we tested the effect of methyl-$\beta$-cyclodextrin towards *Mr*Nvc VLPs internalisation, which is known to inhibit caveolae-mediated endocytosis through cholesterol depletion in cells (*Kim et al., 2012*). Depletion of green granular appearance in

the Sf9 cells further confirmed the importance of caveolae-mediated endocytosis towards the internalisation of *Mr*Nvc VLPs. Methyl-$\beta$-cyclodextrin has also been reported to inhibit clathrin-mediated endocytosis (*Rodal et al., 1999*). To verify the involvement of clathrin-mediated endocytosis, we studied the effect of CPZ, a clathrin-mediated endocytosis inhibitor which inhibits Rho GTPase (*Hussain et al., 2011*; *Kim et al., 2012*). Interestingly, the presence of CPZ halted the *Mr*Nvc VLPs internalisation into Sf9 cells, suggesting that clathrin-mediated endocytosis also plays a role in the VLPs entry into Sf9 cells. To assess the involvement of macropinocytosis, cytochalasin D which inhibits actin polymerisation was used in this study (*Gold et al., 2010*; *Iversen, Skotland & Sandvig, 2011*). From the results, no noticeable inhibition of VLP entry was observed, suggesting that the entry mechanism is macropinocytosis-independent.

Following the attachment of *Mr*Nvc VLPs on the cell surface and the requirement of an acidic environment in early VLP entry, the formation of endosomes and endosomal escape of VLPs were observed via a real-time live cell imaging system (Video S1). The early step for endosome formation is triggered by the attachment of VLPs on Sf9 cell surface. VLPs were brought closer surrounding a membrane pit. The VLPs were later accumulated into the centre of the pit. After a while, the VLPs were observed moving around freely in the cytoplasm in a circular vesicle, which is believed to be an endosome. A disproportionate size and shape of the endosome was observed, followed by fading and disappearance of the fluorescence signal, which could indicate endosomal escape of the *Mr*Nvc (*Ohtsuki et al., 2015*). During endosomal escape, *Mr*Nvc was released into the cytosol. *Kalia & Jameel (2011)* reported that virus particles stay in an endocytic organelle until the conditions permit them to be released or when an endosome is close to the nuclear pore for translocation into a nucleus. Since nodavirus is a non-envelope virus, membrane fusion is unlikely to be involved in the endosomal escape.

*Odegard et al. (2009)* and *Odegard, Banerjee & Johnson (2010)* proposed that the capsid protein of FHV undergoes a conformational change in the acidic environment of endosome and exposes the $\gamma$ peptide (44 residues) at its C-terminal end. This short peptide binds to the endosomal membrane and disrupts the membrane to facilitate translocation of nucleocapsid into the cytoplasm. However in the present study, the $\gamma$ peptide and its cleavage site (for FHV (Asn[363]–Ala[364]; *Odegard, Banerjee & Johnson, 2010*) and Pariacoto virus (Asn[361]–Ser[362]; *Johnson, Zeddam & Ball, 2000*)) are not present at the C-terminal end of *Mr*Nvc based on amino acid sequence analysis. The C-terminal region of *Mr*Nvc localised within the cell nucleus remained intact, as the His-tag at the C-terminal end of the protein was detected by anti-His antibody in Western blotting, suggesting that the *Mr*Nv might use a different endosomal escape mechanism compared to that of FHV.

Many viruses contain a nuclear localisation signal (NLS) comprising basic amino acids which allow them to enter the hosts' nuclei. These viruses include SV40 (*Kalderon et al., 1984*), hepatitis B virus (*Li et al., 2010*), BK polyomavirus (*Bennett et al., 2015*) and dengue virus (*Netsawang et al., 2010*). Since FHV does not enter the host nucleus, GGNNV (a betanodavirus) was used instead to illustrate the possible mechanism employed by *Mr*NV in the nucleus translocation. A highly basic 9-residue peptide ([23]RRRANNRRR[31]) was identified in GGNNV as a nucleus targeting sequence in fish and mammalian cells

(*Guo, Dallmann & Kwang, 2003*). The N-terminal region of *Mr*Nvc is also highly rich in positively-charged residues. A 10-residue peptide, [20]KRRKRSRRNR[29], was identified as the RNA-binding region (*Goh et al., 2014*). This peptide shares some similarities with the NLS of GGNNV. Therefore, the N-terminal deletion mutants of *Mr*Nvc (*Goh et al., 2014*), namely 9Δ*Mr*Nvc, 19Δ*Mr*Nvc, 29Δ*Mr*Nvc and 20–29Δ*Mr*Nvc were used to study their internalisation into Sf9 cells. The nuclear localisation ability of the VLPs reduced dramatically for 20–29Δ*Mr*Nvc mutant. This suggests that the highly basic residues located at residues 20–29 of *Mr*Nvc are part of the nuclear targeting sequence. Amino acid sequence analysis of *Mr*Nvc with the cNLS mapper (*Kosugi et al., 2009*) revealed that the [20]KRRKRSRRNR[29] of *Mr*Nvc is a monopartite NLS with a score of 7.5 (max score of 10; reflects the strength of NLS activities), indicating partial localisation to the nucleus based on an importin-$\alpha$ dependent pathway. These findings suggest that the RNA binding domain of *Mr*Nvc plays a vital role in the nuclear translocation of *Mr*NV. The dual function of RNA binding and nucleus translocation of a highly basic peptide motif has also been reported in other viruses and proteins, such as the Alfafa mosaic virus (*Herranz, Pallas & Aparicio, 2012*) and human dicer (*Doyle et al., 2013*).

## CONCLUSIONS

As a summary, fluorescence microscopy, sub-cellular fractionation and live cell imaging revealed that *Mr*Nvc VLPs were localised in the cytoplasm and nucleus of the Sf9 cells. Upon entry through the clathrin- and caveolae-mediated endocytosis, the *Mr*Nvc was enclosed in endosomes and escaped from this compartment with a mechanism different from FHV. The highly basic RNA-binding domain located at positions 20–29 of the *Mr*Nvc does not play a role in the VLP entry into the cytoplasm, however its function in nuclear translocation was demonstrated. Overall, this study has shed some light on the journey of *Mr*Nvc VLPs in an insect cell, mimicking the native *Mr*Nv.

## ACKNOWLEDGEMENTS

We thank Dr. Ho Kok Lian for his technical advice.

### Funding

This study was funded by the Ministry of Science, Technology and Innovation of Malaysia (MOSTI; Grant no. 02-01-04-SF2115). The funders had no role in study design, data collection and analysis, decision to publish, or preparation of the manuscript.

### Grant Disclosures

The following grant information was disclosed by the authors:
Ministry of Science, Technology and Innovation of Malaysia: 02-01-04-SF2115.

### Competing Interests

The authors declare there are no competing interests.

## Author Contributions

- Ummi Fairuz Hanapi and Chean Yeah Yong conceived and designed the experiments, performed the experiments, analysed the data, wrote the paper, prepared figures and/or tables, reviewed drafts of the paper.
- Zee Hong Goh and Swee Keong Yeap conceived and designed the experiments, analysed the data, wrote the paper, reviewed drafts of the paper.
- Noorjahan Banu Alitheen conceived and designed the experiments, analysed the data, contributed reagents/materials/analysis tools, reviewed drafts of the paper.
- Wen Siang Tan conceived and designed the experiments, analysed the data, contributed reagents/materials/analysis tools, wrote the paper, reviewed drafts of the paper.

## Data Availability

    The raw data are included in the figures.

## Supplemental Information

Supplemental information for this article can be found online at http://dx.doi.org/10.7717/peerj.2947#supplemental-information.

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
