# Peer review of "Tracking the virus-like particles of Macrobrachium rosenbergii nodavirus in insect cells"

_PeerJ, doi:10.7717/peerj.2947_

## Round 0.1 · original submission · Major Revisions

Please, address all the major points addressed by the two reviewers.

Reviewer 1 ·

Basic reporting

The manuscript presented by Hanapi and cols. investigates the entry of a nodavirus that affects prawn industry worldwide, in insect cells using the fluorescent labeling of the capsid proteins as an approach to track Virus-like particles inside the cells. The authors also use the lysosomotropic agent ammonium chloride to investigate if the low pH in endocytic vesicles is important for virus entry and infection. The entry of nodaviruses in different cell types has been investigated by other groups and the data in literature shows that usually they enter by endocytosis and the infection process is dependent on the pH decrease during internalization. The manuscript presented adds information on a nodavirus that seems to be different from the ones investigated before and that was suggested to be classified in a new genus Gammanodavirus. While the gap in current literature is well stated and the manuscript present nice figures with interesting data, there are major points that should be reviewed before considering publication. English is ok for most parts of the text, but it would benefit from a good review for English language as many expressions or phrases used are confusing. An example that can be seen throughout the paper is “VLPs internalize Sf9 cells”. Some of these phrases are highlighted on the annotated pdf file. There is an important lack of information on the experimental design and set up used, mainly for the microscopy experiments (described in the next session) that has to be evaluated. It is not possible to interpret and take conclusions from the data without these details.

Experimental design

The work is mainly based on fluorescence microscopy to track the entry of VLPs in SF9 cells. The labeling and tracking of virus particles has been extensively used as a method to investigate virus-cell interaction. One problem with the approach here in the presented manuscript is that the particles used are not infectious virus particles, but virus-like particles produced by the expression of the capsid protein in bacteria cells. While their interaction with cells can still give some insight about how the virus particles interact with the cell surface and are internalized, there is no infection to be followed. We do not know if the RNA would be regularly released. Particle stability is different and disassembly may happen in different conditions even if the delivery of plasmid DNA has been demonstrated before. Then, all the effects of the ammonium chloride used can not be related to virus replication. The effect observed of reduced number of vesicles is a well known effect of this agent that inhibits vesicle trafficking and fusion.
Authors should supply all details about their experimental set up. There is no information on the objectives, filters, excitation and emission wavelengths used for the fluorescence microscopy experiments. Specifically, under the sub section “Fluorescence microscopy of Sf9 cells incubated with F-MrNvc VLPs” in Materials and Methods, the authors state they used a Leica EZ4 equipment for the fluorescence images with a green filter (?). The Leica EZ4 is a stereomicroscope, in principle, unable to produce fluorescence images. Authors should describe exactly the set up for these experiments.
- The experiments were done in totally different time windows. For the first fluorescence microscopy experiments described in Materials and Methods, VLPs were incubated with cells for 1h at 4o C and then cells were incubated for 16 h (no info on temperature ) while for the live cell experiments they were observed for 1h, after 1h incubation at 4oC and 30 min at room temp in a total time of 1 h 30 min of the infection process, much shorter than the experiments. The 16 h incubation time is used for the ammonium chloride experiments and nuclear localizations studies. Usually, the entry by endocytosis is followed by transport of the particle to the site of RNA release in very early steps of the entry process. Authors should justify the long incubation times in these experiments.

- in Fig 2 B and C, The authors present z-stack images of infected cells but do not describe the equipments or the set up of the experiments. No details on objectives, pinhole size, laser lines and emission filters are given. Neither Leica EZ4 nor Olympus IX81 (described for live cell image) inverted fluorescence microscope are able to produce confocal images of single cells. Only with a confocal fluorescence image set up this kind of analyses could be done. A clear description of the set up and experiment procedure should be provided

- on Effect of ammonium chloride (NH4Cl) on the entry of MrNvc VLPs into Sf9 cells. The authors do not test the cell viability in the presence of NH4CL. Authors incubated with ammonium chloride for 1h and then with virus for 16h. It is not clear if ammonium is present during the entire 16h period. The control of cell viability is even more important for this long incubation time. On the other hand, It is well known that lysosomotropic compounds have a reversible effect and endosome and lysosome pH decreases shortly after washing the drug out.

Validity of the findings

The results clearly show that VLPs are able to interact with the cells and are internalized in vesicles inside the cytoplasm of the host cells. Unfortunately, the description of the experimental set up does not allow a clear evaluation of the conclusion that viral proteins are found inside the nuclei of infected cells. The authors present z-stacks of an infected cell. No details on the experimental set up are given. No objective, pinhole size, laser lines used, emission wavelengths collected. The microscope IX81 described by the authors is an epi-fluorescence microscope and is not able to produce confocal fluorescence images, if this is the equipment configuration. In this case, this is not an appropriate experiment to show specific localization of the VLPs or virus proteins in the z direction, as inside or outside of the nucleus.
In figure 4, the authors state to observe a coated pit surrounded by virus particles, that are then brought together at the center of the pit and internalized. The maximum average size expected for a coated pit is about 200 nm. This is close to the best resolution limit in an optical microscope. It would be very hard to observe something like a ring shape with this diameter formed by particles of 30 nm, that would show as diffraction limited spots of 200 nm in an optical microscope working in its best performance. Besides all that, The scale bar on the figures are 5 um, as stated by the authors. A simple direct comparison of the size of ring shaped structure and the bars strongly suggests that the structure dimension is about 3 um. The authors can not say that this is a coated pit surrounded by VLPs. This is either some other structure, or it is an artifact of the experiment. It is possible that particles are engulfed by pinocytosis, for example.
The authors seem to consider that the green fluorescence comes from VLPs even after the 16 h of the internalization process, and the supposedly nuclear localized fluorescence observed. There is no evidence that these are particles. They could be viral proteins from disassembled particles. If the authors believe these are particles they should clearly state this as speculation and explain why they believe it.
When authors discuss the entry machanisms they say they can observe the release of particles into the cytoplasm. There is no evidence for that. They also say that the particles are internalized in endosomes that evolve to late endosomes. There is no evidence for that as there is no markers that give any evidence of endosomes or late endosomes. This speculative.
Next, authors say that the “rates of endosome formation and endosomal escape were inconsistent”. No data was presented on the rate of formation or escape from endosomes at any part of the paper. Where this information comes from?

Additional comments

no comments

Annotated reviews are not available for download in order to protect the identity of reviewers who chose to remain anonymous.

·

Basic reporting

While the structure of the paper is generally fine there are many awkward sentences and missing or inappropriately used words. Hence the manuscript would benefit from professional language editing.
The paper feels incremental. Some of the conclusions are speculative, example being the claim that the results suggest Sf9 (insect) cells reproduce the infection in the natural host (prawn). Many viruses would enter heterologous cells but often via different, non-specific mechanisms. Authors provide more background information on the type of receptor or better yet infect Sf9 cells with the virus.
Merge color figures in Fig. 1C and Fig. 5C shall be enlarged to show localization to the nucleus more clearly.

Experimental design

There are several issues with appropriate controls (absence of) in the experimental design:
1) I have doubts that the cell fractionation yielded pure nuclear fraction. In order to show that assays or Western blot for cytoplasmic, endosomal and nuclear markers shall be performed on both fractions to show their purity. I suspect the nuclear fraction contains also endosomes.
2) Western blots should contain control lanes with purified VLPs
3) How much VLPs resemble the mature virus? There is effect of viral genomic RNA on the local stability of the viral particle, as demonstrated by Bothner et al 1998
JBC 273, 673-676 and others. Are these VLPs void of RNA?
4) Disappearance of fluorescence spots in life cell imaging could be a simple result of dye photobleaching which for fluorescein is a considerable problem. Transfer from one compartment to another is usually demonstrated by tracking the particles. I would recommend to use more stable dye, like atto488 or Alexa488.

Validity of the findings

Many of the findings are compromised by the flaws in the experimental design as indicated above so unless adequate controls are introduced the conclusions are very speculative or could be even wrong.

Additional comments

This is an interesting paper so it would be nice to see the results further substantiated by better controls and more live cell imaging.

---

## Round 0.2 · accepted · Accept

The authors have properly revised the manuscript.